# Botulinum Toxin Type-A (Botulax^®^) Treatment in Patients with Intractable Chronic Occipital Neuralgia: A Pilot Study

**DOI:** 10.3390/toxins13050332

**Published:** 2021-05-04

**Authors:** Hyeyun Kim, Bongjin Jang, Seong-Taek Kim

**Affiliations:** 1Department of Neurology, International St. Mary’s Hospital, College of Medicine, Catholic Kwandong University, 25, 100 Beon-gil, Simgok-ro, Seo-gu, Incheon 22711, Korea; imkhy77@gmail.com; 2Department of Medical Business Administration, Daegu Haany University, 201, Daegudae-ro, Gyeongsan-si 38610, Korea; jangbj@gmail.com; 3Department of Orofacial Pain and Oral Medicine, Yonsei University College of Dentistry, 50-1 Yonsei-ro, Seodaemun-gu, Seoul 03722, Korea

**Keywords:** chronic occipital neuralgia, greater occipital nerve, botulinum toxin injection

## Abstract

Intractable chronic occipital neuralgia (ON) is an uncommon type of headache often experienced by patients in outpatient neurological clinics. Among patients unresponsive to oral neuralgia medications, needling or injections with several drugs were suggested alternatives for treating chronic ON. This study aimed to determine the effectiveness and safety of botulinum toxin type-A (BTX-A) injection treatments, where eight patients with unilateral chronic ON received BTX-A injections at the pain sites. The pain relief effect was observed 2 weeks after receiving the injections, gradually showing improvements up to 12 weeks after injection. There were no adverse events or changes from baseline in serologic studies and vital signs in any of the participants. The treatment’s pain-relieving effects were confirmed through regular, 12-week follow-ups, confirming the safety and effectiveness of BTX-A on chronic ON and suggesting that this method is an effective, novel alternative option for chronic ON treatment.

## 1. Introduction

Chronic occipital neuralgia (ON) is a refractory headache characterized by sudden, sharp, and pounding pain in the branches of the occipital nerves. Specifically, ON can be described as bilateral or unilateral, typically accompanied by paresthesia around the occipital nerve [1]. During the initial, mild stage, this headache is commonly treated with nonsteroidal anti-inflammatory drugs (NSAIDs), tricyclic antidepressants, or anticonvulsants [2,3]. If the patient’s condition does not improve after using those medications, trigger point injection, physical therapy, pulsed radiofrequency therapy, occipital nerve surgical decompression, and botulinum toxin type-A (BTX-A) injection on target sites could be considered [4,5,6,7]. However, there are no clear treatment guidelines for chronic ON patients who do not respond to medical treatment. This study aimed to determine the efficacy and safety of BTX-A injections for use in chronic ON therapy.

## 2. Results

After screening and providing their informed consent, nine patients participated in this study. However, one patient was excluded after a baseline serologic examination showed a hemoglobin level of 9 g/dL, reducing the total number of participants to eight individuals (three male, five female) with an average age of 46.3 ± 7.8 years. The pre-clinical serologic findings did not show any notable findings in the study participants, and no differences in the serologic findings before and after injecting the BTX-A were observed.

The primary efficacy variable was the change in the pain visual analog scale (VAS), evaluated by the investigator before and thrice on the 2nd, 4th, and 12th week after administration. Based on the clinical trial’s results, pain relief was continuously observed after the 2nd week, showing statistical significance (*p* = 0.011, *F* = 125.49; Figure 1), and the most notable change was observed 2 weeks after injection (Table 1). After injecting 20 U of BTX-A, two patients took the pain killer intermittently for up to 4 weeks whenever they were in pain, but the rest of the participants experienced improved laryngeal pain after 2 weeks and did not need to take painkillers anymore. There was no significant change in the quality of life (QOL) evaluation throughout the study, and the participants did not experience adverse reactions before or after BTX-A treatment. In addition, there were no significant differences in the posttreatment hematological results, which were similar to the serologic results obtained before BTX-A treatment.

## 3. Discussion

Chronic ON is a painful disease that occurs in the occipital and cervical spine, where the occipital nerve in the occipital region’s second cervical spine is compressed or if inflammation is present [4]. The pain’s frequency, intensity, and pattern can vary, ranging from depressed without sensation to a continuous, tingling feeling along the innervation area [1,4]. Medical treatment with oral painkillers is usually performed, but if pain persists, invasive treatment options, including dry needling or acupuncture, are used to induce immediate pain relief [3]. Pain relief can last for about 2 weeks using dry needling or acupuncture [8,9,10]. In this study, pain relief observed within the first two weeks was similar to the reported effects of dry needling or acupuncture. With BTX-A injection, pain relief was observed after the first two weeks [11]. However, because BTX-A was injected at a different site compared to dry needling or acupuncture, directly comparing the three methods is difficult.

As observed by patients, pain relief is only experienced briefly by patients who underwent acupuncture or dry needling, and pain often recurs. When the pain is not alleviated using different methods, compression was discovered to be the cause, and the occipital nerve would be surgically amputated. Although botulinum toxin injection treatment is an uncommon treatment option, several clinical case studies have been performed to ascertain its effectiveness. Similarly, this study was designed to demonstrate this treatment option’s effectiveness by injecting botulinum toxin into patients with chronic laryngeal neuralgia. As a result, clinical symptoms started improving after 2 weeks in all participants and improved continuously over 4 and 12 weeks after injection.

In a previous study, BTX-A for chronic ON was injected in five patients with bilateral ON and one patient with unilateral ON [12]. In two previous case series involving six patients with chronic ON, it was reported that BTX-A effectively relieves pain. Compared to a previous study wherein 50 U of BTX-A was injected for each pain area, this study administered 20 U of BTX-A per pain site, while providing the detailed description of the injection site [12,13]. The comparison with the previous two case series is summarized in Table 2. In each patient with ON, 50 U of botulinum toxin was diluted in 3 mL of 0.9% normal saline solution and injected into the branches of the large and small laryngeal nerves. As a result, chronic ON was significantly reduced in the post-injection VAS from the baseline VAS. In addition, the overall QOL improved, and the use of analgesics also decreased. In this study, all patients were diagnosed with unilateral chronic ON, and the pain-relieving effects of the BTX-A treatment were positive. However, changes in each patient’s QOL after BTX-A treatment were not observed.

Theoretically, a neuromuscular junction blocking substance, such as botulinum toxin, would block the secretion of acetylcholine and suppress pain [14]. In addition, BTX-A has also been shown to effectively alleviate neuropathic pain in several animal studies. Pain relief for neuralgia occurred because BTX-A inhibited the secretion of neurotransmitters involved in pain mechanisms (protein P mediators or the nerve ending and dorsal root ganglia’s glutamate and calcitonin genes) or processes (reducing inflammation around nerve endings or deactivating sodium channels) [14,15].

Adverse treatment reactions were investigated to evaluate the safety of botulinum toxin injections, and follow-up tests were performed to obtain the participants’ vital signs and serologic findings. No adverse reactions were reported in all participants, and no changes in the participants’ vital signs and serological results were observed after toxin injection compared with baseline results.

This study demonstrated the effectiveness of botulinum toxin injection treatment in chronic ON patients whose conditions did not improve with oral medication. In addition, significant pain relief was noted 2 weeks after BTX-A injection, and pain intensity gradually decreased up to 12 weeks after treatment. Chronic ON is a rare disease that lacks a well-designed randomized clinical trial. However, there are clinical series of treatment options other than drug therapy. There are also no comparative studies between pharmacologic and non-pharmacologic treatments. Based on the results of previous studies, BTX-A treatment is less invasive, has a rapid onset of drug effects, and maintains the effect of pain reduction for a relatively long time [16].

The study’s main limitation lies in the small number of participants because chronic ON is a low-prevalence disease. Therefore, conducting a large-scale clinical study in a single institution is difficult, especially with a limited study time. In addition, large-scale, multicenter studies are needed to establish BTX-A injection treatment procedures for chronic ON. Future studies can also compare the BTX-A injection method’s effectiveness to dry needling using the latter’s injection sites.

## 4. Conclusions

This study determined that BTX-A injection could be considered an alternative treatment option for chronic ON patients unresponsive to oral drug treatment. The treatment’s pain-relieving effects were observed 2 weeks after injection and sustained for up to 12 weeks. Based on the study’s limitations, future large-scale studies should establish a basis for this treatment option.

## 5. Materials and Methods

### 5.1. Subjects

The participants recruited for this study were adult men and women over 20 years old and patients with ON who met the International Classification of Headache Disorder (ICHD-3) diagnostic criteria. All participants fully understood the aims of this study and voluntarily agreed. This study was conducted following the Declaration of Helsinki, and the protocols were approved by the appropriate ethics review board (#IS19MIST0008). All participants gave their consent before participating in the study.

However, this study contained more extensive exclusion criteria. In general, those who received BTX-A injections into the occipital region within the year; those whose masses were touched in the occipital region; and those diagnosed with severe muscular asthenia, Eaton-Lamberton syndrome, amyotrophic lateral sclerosis, motor neuropathy, or similar diseases were excluded. In addition, participants who had taken aminoglycoside antibiotics, curare-like agents, or drugs that inhibit neuromuscular functions (muscle relaxants, anticholinergics, benzodiazepines, benzamides, tetracycline, lincomycin antibiotics) four weeks before screening or who were allergic or sensitive to investigational drugs or their ingredients were also excluded. Among women, those who were pregnant, lactating, likely pregnant, or who did not agree to use medically accepted contraception during the trial period were excluded from the study as well. The study’s investigators also excluded those whom they deemed unsuitable for this clinical trial. These included patients who participated in other clinical trials up to 30 days before screening or took drugs that had not exceeded 5 times their half-life during the clinical trial.

### 5.2. Occipital Neuralgia Diagnostic Criteria

According to the International Classification of Headache Disorders (ICHD-3), ON belongs to the same family as cranial neuralgias, central and primary facial pain, and other headaches. The diagnostic criteria include paroxysmal stabbing pain, with or without persistent aching between paroxysms, in the distribution of the greater, lesser, or third occipital nerve, tenderness over the affected nerve, and pain that was eased temporarily by locally blocking the nerve with anesthetics.

### 5.3. Procedure

Preservative-free sterile physiological saline (0.9% sodium chloride injection) was first used to dissolve the lyophilized investigational drug. Next, Botulax^®^ single bottle of 50 U was diluted in 3 mL of 0.9% physiological saline. Lastly, a 27 G needle was inserted into the patient until the end of the bone touched the occipital ridge’s posterior along the nape at a 90° angle toward the occipital region (Figure 2). Injection occurred along the intersection of the greater and lesser occipital nerves.

The great occipital nerve (GON) is located at the third point of the virtual line connecting the external occipital protuberance (inion) and the mastoid process. The injection sites are as follows: at a third of the virtual line connecting the mastoid process and inion, a point in the shape of an inverted triangle set around 3 cm from the first site, and the other 2 points are the vertices of the inverted triangle. Finally, BTX-A was injected into 4 points (red dots) around GON, where 5 U of BTX-A was administered for each point (Figure 2).

### 5.4. Statistical Analysis

All values obtained in this study are presented as mean ± standard deviations. The difference between the mean values of the VAS and QOL baseline variables and the 2-week, 4-week, and 12-week treatments were analyzed using repeated measures ANOVA. A *p*-value of less than 0.05 was considered statistically significant. All analyses were conducted using IBM Statistics SPSS (version 26.0 SPSS, Inc., Chicago, IL, USA).

## Figures and Tables

**Figure 1 toxins-13-00332-f001:**
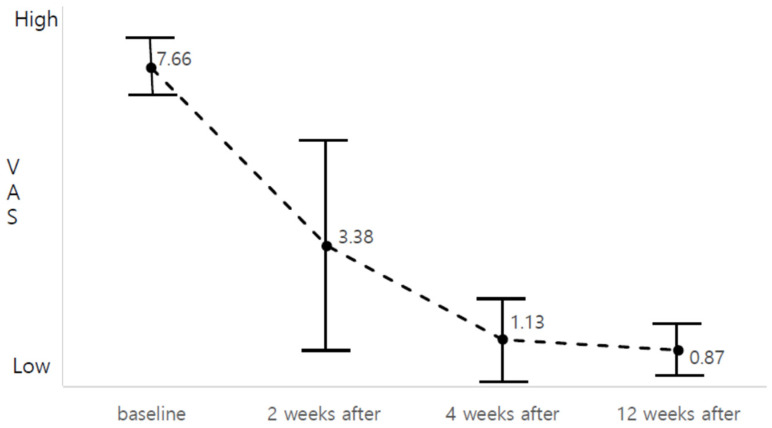
Changes in the pain visual analog scale (VAS) after botulinum toxin (BTX) injection. The pain relief was continuously observed from the 2nd week to 12th week after injection, and the most notable change was observed 2 weeks after injection.

**Figure 2 toxins-13-00332-f002:**
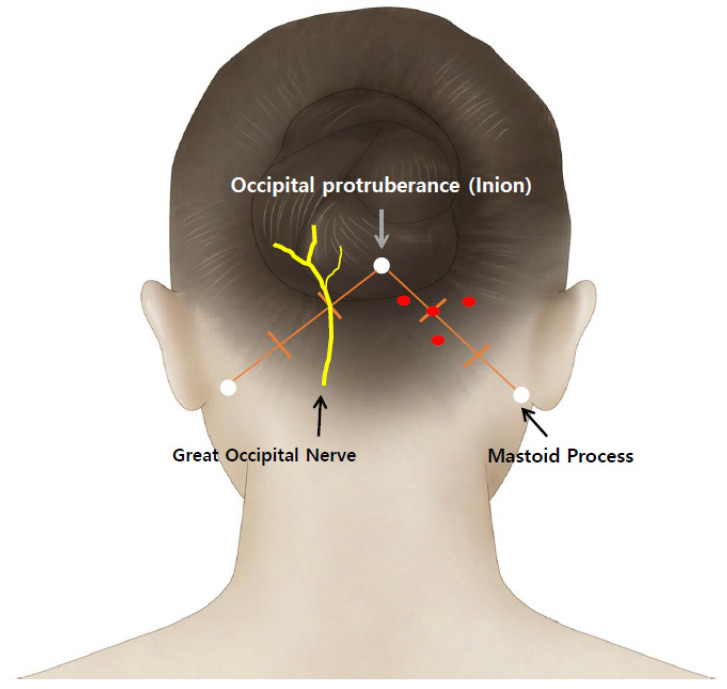
The injection site of BTX-A.

**Table 1 toxins-13-00332-t001:** Changes in pain visual analog scale (VAS) and quality of life (QOL) after treatment.

Variables	Baseline	2 Weeks after Treatment	4 Weeks after Treatment	12 Weeks after Treatment	*F*
VAS	7.66 ± 0.70	3.38 ± 2.50 ***	1.13 ± 0.99 ***	0.87 ± 0.64	125.49 ***
QOL	76.67 ± 9.95	70.88 ± 8.22	70.38 ± 4.41	72.0 ± 4.93	997.15

*** *p* = 0.000.

**Table 2 toxins-13-00332-t002:** Comparison of previous studies with BTX-A treatment for chronic ON.

Variables	Number of Cases	Total Unit of BTX-A	Median Score Differences of VAS	Median Score Differences of 2nd Outcome	Duration of Pain Relief (Weeks)
Kapural	6	50 U	7.5	34 ^1^	16
Taylor	6	50 U	5.8	61 ^2^	6
Kim	8	20 U	6.5	NC ^3^	4

VAS, visual analog scale; ^1^ PDI, Pain Disability Index; ^2^ HSQL, Headache Specific Quality of Life; ^3^ QOL, quality of life.

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
