# Peer review of "Botulinum Toxin Type-A (Botulax®) Treatment in Patients with Intractable Chronic Occipital Neuralgia: A Pilot Study"

_toxins, 2021, doi:10.3390/toxins13050332_

Round 1
Reviewer 1 Report
The interesting study deals with an infrequent pathology that is difficult to treat and diagnose.
There are some concerns to be evaluated.
- As acknowledged by the authors, the small number of cases does not allow definitive conclusions to be drawn. A sentence that remembers this aspect should be included in the conclusions section
- It should be clarified how many infiltrations are made and how many IU are administered.
- As the authors explain, there is no improvement in the Quality of Life due to a clear reduction in pain.
- It would be necessary to insert a table with the consumption of analgesics before and after treatment.
- To have a scientifically relevant result, the study should be carried out with a placebo arm.
- The references should be completed (attached some examples)
Some recent references (as examples)
- Botulinum Neurotoxin for the Treatment of Neuropathic Pain. Egeo G, Fofi L, Barbanti P. Front Neurol. 2020 Aug 11;11:716. doi: 10.3389/fneur.2020.00716. eCollection 2020. PMID: 32849195 Free PMC article. Review.
- A Review of the Recent Findings in Minimally Invasive Treatment Options for the Management of Occipital Neuralgia. Urits I, Schwartz RH, Patel P, Zeien J, Connor D, Hasoon J, Berger AA, Kassem H, Manchikanti L, Kaye AD, Viswanath O. Neurol Ther. 2020 Dec;9(2):229-241. doi: 10.1007/s40120-020-00197-1. Epub 2020 Jun 2. PMID: 32488840 Free PMC article. Review.
- Botulinum Toxin Type A for refractory trigeminal neuralgia in older patients: a better therapeutic effect. Wu S, Lian Y, Zhang H, Chen Y, Wu C, Li S, Zheng Y, Wang Y, Cheng W, Huang Z. J Pain Res. 2019 Jul 17;12:2177-2186. doi: 10.2147/JPR.S205467. eCollection 2019. PMID: 31410051 Free PMC article.
- Occipital Neuralgia and Cervicogenic Headache: Diagnosis and Management. Barmherzig R, Kingston W. Curr Neurol Neurosci Rep. 2019 Mar 19;19(5):20. doi: 10.1007/s11910-019-0937-8. PMID: 30888540 Review.
- Occipital Neuralgia: a noninvasive therapeutic approach. López-Soto PJ, Bretones-García JM, Arroyo-García V, García-Ruiz M, Sánchez-Ossorio E, Rodríguez-Borrego MA. Rev Lat Am Enfermagem. 2018 Nov 14;26:e3067. doi: 10.1590/1518-8345.2621.3067. PMID: 30462782 Free PMC article.
Author Response
Reviewer 1
- As acknowledged by the authors, the small number of cases does not allow definitive conclusions to be drawn. A sentence that remembers this aspect should be included in the conclusions section
à Thank you for the recommendation. A narrative was added to do not draw a definitive conclusion. It has been modified wherein the revised text is marked in blue (line 130).
“Conclusion
This study determined that BTX-A injection could be considered an alternative treatment option for chronic ON patients unresponsive to oral drug treatment. The treatment’s pain-relieving effects were observed 2 weeks after injection and sustained for up to 12 weeks. Based on the study’s limitations, future large-scale studies should establish a basis for this treatment option.”
- It should be clarified how many infiltrations are made and how many IU are administered.
à The Procedure subsection under the Materials and Methods has been modified as follows. (lines 172–180)
Figure 2. The injection site of BTX-A.
“The great occipital nerve (GON) is located at the third point of the virtual line connecting the external occipital protuberance (inion) and the mastoid process. The injection sites are as follows: at a third of the virtual line connecting the mastoid process and inion, a point in the shape of an inverted triangle around 3 cm from the first site, and the 2 other points are the vertices of the inverted triangle. Finally, BTX-A was injected into 4 points (red dots) around GON, where 5 U of BTX-A was administered for each point (Figure 2).”
- As the authors explain, there is no improvement in the Quality of Life due to a clear reduction in pain.
à There are different factors aside from pain that can affect the quality of life (QOL). However, we considered that QOL could be affected if the pain intensity was enough to create difficulties in everyday life. Based on Table 1, the QOL score did not show a considerable amount of change. This result suggests that the patient’s pain level was not clearly related to the quality of life.
- It would be necessary to insert a table with the consumption of analgesics before and after treatment.
à We added the following sentence in blue to the Results section to discuss the participants’ analgesics consumption. (lines 50–53)
“The primary efficacy variable is the change in the pain visual analog scale (VAS), evaluated by the investigator before and thrice on the 2nd, 4th, and 12th week after administration. Based on the clinical trial’s results, pain relief was continuously observed after the 2nd week, showing statistical significance (p = 0.011, F = 125.49; Figure 1), and the most notable change was observed 2 weeks after injection (Table 1). After injecting 20U of BTX-A, 2 patients took the pain killer intermittently for up to 4 weeks whenever they were in pain, but the rest of the participants experienced improved laryngeal pain after 2 weeks and did not need to take painkillers anymore.”
- To have a scientifically relevant result, the study should be carried out with a placebo arm.
à We acknowledge the reviewer’s opinion. However, as mentioned above, the prevalence of this disease was low, making it difficult to perform the study by enrolling patients for a limited time. Thus, the single-arm study was conducted. The researchers plan to expand patient recruitment through multicenter research and verify this treatment’s effectiveness through placebo control studies.
- The references should be completed (attached some examples)
à We have added and organized the references according to your recommendations.
- Barmherzig, R.; Kingston, W. Occipital neuralgia and cervicogenic headache: diagnosis and management. Curr Neurol Neurosci Rep. 2019, 19, 20. https://doi.org/10.1007/s11910-019-0937-8
- Urits, I.; Schwartz, R.H.; Patel, P.; Zeien, J.; Connor, D.; Hasoon, J.; Berger, A.A.; Kassem, H.; Manchikanti, L.; Kaye, A.D.; Viswanath, O. A review of the recent findings in minimally invasive treatment options for the management of occipital neuralgia. Neurol Ther. 2020, 9, 229–241. https://doi.org/10.1007/s40120-020-00197-1
- López-Soto, P.J.; Bretones-García, J.M.; Arroyo-García, V.; García-Ruiz, M.; Sánchez-Ossorio, E.; Rodríguez-Borrego, M.A. Occipital neuralgia: a noninvasive therapeutic approach. Rev Lat Am Enfermagem. 2018, 26, e3067. https://dx.doi.org/10.1590%2F1518-8345.2621.3067
- Egeo, G.; Fofi, L.; Barbanti, P. Botulinum neurotoxin for the treatment of neuropathic pain. Front Neurol. 2020, 11, 716. https://doi.org/10.3389/fneur.2020.00716
- Wu, S.; Lian, Y.; Zhang, H.; Chen, Y.; Wu, C.; Li, S.; Zheng, Y.; Wang, Y.; Cheng, W.; Huang, Z. Botulinum toxin type A for refractory trigeminal neuralgia in older patients: a better therapeutic effect. J Pain Res. 2019, 12, 2177–2186. https://doi.org/10.2147/JPR.S205467

Reviewer 2 Report
The authors used Botulinum toxin (Botox) to treat patients with occipital neuralgia (ON) in this clinical study. They found that patient's pain can be relieved by Botox injection at the occipital pain sites from 2 to 12 weeks after local Botox injections. This is an article related to daily clinical practice. However, the major concern is that two similar reports have discussed the same topic (Kapural L, 2007[1]; Taylor M, 2008[2]). Furthermore, those two reports had similar results to this study. The author had a little discussion about Taylor M's study in the manuscript (page 3, line 77-82). However, those discussions didn't mention the novelty of this study. I think that the authors can have more discussions about the novelty of this study and the differences between this study and the previous two reports (ex: recruit more patients, change the dose or interval of Botox treatment, arrange a randomized control study).
There are some other minor suggestions to each part of the article:
- The authors can have more discussion about the current guideline of standard ON treatment and other alternative treatments in the introduction (ex: Botox injection, pulsed radiofrequency treatment, occipital nerve surgical decompression). The author can cite the previous two reports about Botox treatment of ON, and discussion the shortage of those two reports.
- The authors can have more discussion about the advantages and benefits of Botox in clinical ON treatment compared to other alternative treatments. (ex: acupuncture, pulsed radiofrequency treatments, occipital nerve surgical decompression)
Reference:
- Kapural, L.; Stillman, M.; Kapural, M.; McIntyre, P.; Guirgius, M.; Mekhail, N. Botulinum toxin occipital nerve block for the treatment of severe occipital neuralgia: a case series. Pain Pract 2007, 7, 337-340, doi:10.1111/j.1533-2500.2007.00150.x.
- Taylor, M.; Silva, S.; Cottrell, C. Botulinum toxin type-A (BOTOX) in the treatment of occipital neuralgia: a pilot study. Headache 2008, 48, 1476-1481, doi:10.1111/j.1526-4610.2008.01089.x.
Author Response
- The authors used Botulinum toxin (Botox) to treat patients with occipital neuralgia (ON) in this clinical study. They found that patient’s pain can be relieved by Botox injection at the occipital pain sites from 2 to 12 weeks after local Botox injections. This is an article related to daily clinical practice. However, the major concern is that two similar reports have discussed the same topic (Kapural L, 2007[1]; Taylor M, 2008[2]). Furthermore, those two reports had similar results to this study. The author had a little discussion about Taylor M’s study in the manuscript (page 3, line 77-82). However, those discussions didn’t mention the novelty of this study. I think that the authors can have more discussions about the novelty of this study and the differences between this study and the previous two reports (ex: recruit more patients, change the dose or interval of Botox treatment, arrange a randomized control study).
à Thanks for your recommendation. The differences with existing research are described in detail in the Discussion section.
“In a previous study, BTX-A for chronic ON was injected in five patients with bilateral ON and one patient with unilateral ON [12]. In 2 previous case series involving 6 patients with chronic ON, it was reported that BTX-A effectively relieves pain. Compared to the previous study wherein 50U of BTX-A was injected for each pain area, this study administered 20U of BTX-A per pain site, while providing the detailed description of the injection site [12,13].” (lines 91–95)
- There are some other minor suggestions to each part of the article:
- The authors can have more discussion about the current guideline of standard ON treatment and other alternative treatments in the introduction (ex: Botox injection, pulsed radiofrequency treatment, occipital nerve surgical decompression). The author can cite the previous two reports about Botox treatment of ON, and discussion the shortage of those two reports.
à Thank you for your suggestion. We added the currently existing technologies for ON treatment in the Introduction.
“If the patient’s condition does not improve after using those medications, trigger point injection, physical therapy, pulsed radiofrequency therapy, occipital nerve surgical decompression, and botulinum toxin type-A (BTX-A) injection on target sites could be considered [4–7].” (lines 34–35)
- The authors can have more discussion about the advantages and benefits of Botox in clinical ON treatment compared to other alternative treatments. (ex: acupuncture, pulsed radiofrequency treatments, occipital nerve surgical decompression)
à The sentences in blue are added to the Discussion section following the reviewer’s recommendation.
“This study demonstrated the effectiveness of botulinum toxin injection treatment in chronic ON patients whose conditions did not improve with oral medication. In addition, significant pain relief was noted 2 weeks after BTX-A injection, and pain intensity gradually decreased up to 12 weeks after treatment. Chronic ON is a rare disease that lacks a well-designed randomized clinical trial. However, there are clinical series of treatment options other than drug therapy. There are also no comparative studies between pharmacologic and non-pharmacologic treatments. Based on the results of previous studies, BTX-A treatment is less invasive, has a rapid onset of drug effects, and maintains the effect of pain reduction for a relatively long time.” (lines 117–122)
à The following references are added.
- Taylor, M.; Silva, S.; Cottrell, C. Botulinum toxin type-A (BOTOX) in the treatment of occipital neuralgia: a pilot study. Headache. 2008, 48, 1476–1481. https://doi.org/ 1111/j.1526-4610.2008.01089.x
- Kapural, L.; Stillman, M.; Kapural, M.; McIntyre, P.; Guirgius, M.; Mekhail, N. Botulinum toxin occipital nerve block for the treatment of severe occipital neuralgia: a case series. Pain Pract. 2007, 7, 337–340. https://doi.org/10.1111/j.1533-2500.2007.00150.x
Round 2
Reviewer 1 Report
In this form the article is suitable to be published in Toxins Journal
Author Response
April 26, 2021
Prof. Dr. Jay Fox
Editor-in-Chief
Toxins
Dear Dr. Fox:
We would like to thank you and the reviewers for your time and constructive feedback in improving our manuscript entitled “Botulinum Toxin Type‐A (Botulax®) Treatment in Patients with Intractable Chronic Occipital Neuralgia: A Pilot Study” (16 references), which we submitted to Toxins. As requested, we include this response in the decision letter. Please see our responses to reviewer 2’s comments after this letter.
We hope that the changes we made have significantly improved the quality of our manuscript. Please do let us know if there is anything else we can do at this stage. We would be more than happy to do everything we can to assist you in this process.
We look forward to hearing from you!
Reviewer 2 Report
In this revision, the authors concluded that the current study used less Botox than the previous study (20U vs. 50U), and the current study provided more detailed descriptions of the injection site with the picture. They also added more descriptions of standard and alternative treatments for occipital neuralgia (ON).
There are some minor suggestions:
- The author may consider making a table that displays and compares the patients’ characters (ex: unilateral/bilateral ON, age, et al.), methods (ex: 20U vs. 50U), and the treatment effects of those three studies of Botox in clinical ON treatment (Current study; Kapural L, 2007; Taylor M, 2008). In discussion, the authors described “In each patient with ON, 50U of botulinum toxin was diluted in 3 ml of 0.9% 85 normal saline solution and injected into the branches of the large and small laryngeal nerves. As a result, chronic ON was significantly reduced in the post-injection VAS than the baseline VAS. In addition, the overall QOL improved, and the use of analgesics also decreased” (page 3, line 85-88). Did those sentences describe Taylor’s study (Taylor M, 2008)? Please clarify and make the comparisons in the table or the discussion.
- In the abstract, the authors said those eight patients had unilateral chronic ON. However, the patients’ ON characters (ex: unilateral or bilateral) were not described in detail in methods and material. In methods and material, the author said 50U Botox was used to treat patients (page 4, line 156), and 5U of BTX-A was administered for each point (page 5, line 169). Please clarify in detail (ex: total 50U or 20U?). The description in results “After injecting 20U of BTX-A (page 1, line 42)” may be deleted if the authors have detailed Botox dose descriptions in methods and material.
- The authors used “laryngeal pain (page 1, line 44)”. Did this term indicate cervical or neck pain? Did the descriptions of the “large and small laryngeal nerve (page 3, line 86)” indicate “greater and lesser branches of the occipital nerve”? Did the term “laryngeal neuralgia (page 3, line 77-78)” indicate occipital neuralgia. Please clarify.
